# Wbm0076, a candidate effector protein of the *Wolbachia* endosymbiont of *Brugia malayi*, disrupts eukaryotic actin dynamics

**Michael K. Mills**[1], **Lindsey G. McCabe**[1], **Eugenie M. Rodrigue**[1], **Karl F. Lechtreck**[2], **Vincent J. Starai**[1,3] *

**1** Department of Microbiology, University of Georgia, Athens, Georgia, United States of America,
**2** Department of Cellular Biology, University of Georgia, Athens, Georgia, United States of America,
**3** Department of Infectious Diseases, University of Georgia, Athens, Georgia, United States of America

* vjstarai@uga.edu

## Abstract

*Brugia malayi*, a parasitic roundworm of humans, is colonized by the obligate intracellular bacterium, *Wolbachia pipientis*. The symbiosis between this nematode and bacterium is essential for nematode reproduction and long-term survival in a human host. Therefore, identifying molecular mechanisms required by *Wolbachia* to persist in and colonize *B. malayi* tissues will provide new essential information regarding the basic biology of this endosymbiosis. *Wolbachia* utilize a Type IV secretion system to translocate so-called "effector" proteins into the cytosol of *B. malayi* cells to promote colonization of the eukaryotic host. However, the characterization of these *Wolbachia* secreted proteins has remained elusive due to the genetic intractability of both organisms. Strikingly, expression of the candidate *Wolbachia* Type IV-secreted effector protein, Wbm0076, in the surrogate eukaryotic cell model, *Saccharomyces cerevisiae*, resulted in the disruption of the yeast actin cytoskeleton and inhibition of endocytosis. Genetic analyses show that Wbm0076 is a member of the family of Wiskott-Aldrich syndrome proteins (WAS [p]), a well-conserved eukaryotic protein family required for the organization of actin skeletal structures. Thus, Wbm0076 likely plays a central role in the active cell-to-cell movement of *Wolbachia* throughout *B. malayi* tissues during nematode development. As most *Wolbachia* isolates sequenced to date encode at least partial orthologs of *w*Bm0076, we find it likely that the ability of *Wolbachia* to directly manipulate host actin dynamics is an essential requirement of all *Wolbachia* endosymbioses, independent of host cell species.

## Author summary

Filarial nematodes of the family Onchocercidae cause several debilitating human diseases such as lymphatic filariasis and onchocerciasis; more than 50 million people are infected by these arthropod-borne roundworms in mostly tropical and sub-tropical regions. Many of these nematodes, including *Brugia malayi*, are obligately colonized by an intracellular bacterium of the genus *Wolbachia*, which is absolutely required for the proper

**Data Availability Statement:** All relevant data are within the manuscript and its Supporting Information files.

**Funding:** M.K.M. is an ARCS Foundation Scholar (https://www.arcsfoundation.org/), and V.J.S. is supported in part by a grant from the National Institute of Allergy and Infectious Diseases (R01-AI100913) and the University of Georgia Department of Microbiology. K.F.L. is supported by a grant from the National Institutes of Health (R01-GM110413). The funders had no role in study design, data collection and analysis, decision to publish, or preparation of the manuscript.

**Competing interests:** The authors have no competing interests to declare.

development and reproduction of these worms in a mammalian host. Clearance of *Wolbachia* from these nematodes leads to a loss of both worm viability and its ability to cause disease in humans. Efforts to understand the molecular interactions required to maintain this important bacterium: nematode endosymbiosis, however, have been hampered due to the genetic intractability of these organisms. In this work, we utilize yeast as a surrogate eukaryotic cell to show that a candidate secreted effector protein from *Wolbachia*, Wbm0076, disrupts eukaryotic actin dynamics and endocytosis. We also observe interactions of Wbm0076 with a highly-conserved eukaryotic actin regulatory protein. As some intracellular bacteria manipulate host actin dynamics to promote mobility within or into host cells, our study provides evidence of an important *Wolbachia* protein activity that may be essential for its proper localization during the development of *B. malayi*.

## Introduction

The parasitic filarial nematode *Brugia malayi* is a causative agent of lymphatic filariasis, a devastating and neglected human tropical disease, reported to have infected approximately 120 million individuals world-wide at the turn of the century [1]. These mosquito-borne nematodes are transmitted as infective larvae (L3) to human hosts through a blood meal and adult worms persist for years in untreated individuals (5–14 years), causing advanced disease states (elephantiasis) [2] that place a massive humanitarian burden on countries endemic for the causative organisms [3]. Lymphatic filariasis has been successfully targeted by the World Health Organization (WHO) [4, 5]} through mass drug administration (MDA) in endemic countries [5, 6]}. Unfortunately, despite the documented success of WHO in blocking vector transmission of microfilariae to humans, global elimination of lymphatic filariasis did not occur by the proposed year 2020, and more than 800 million people remain threatened by the disease [4, 5, 7]. Therefore, there remains a need to discover anti-filarial, adulticidal treatments to support the long war against human filarial diseases [4, 8, 9].

In the twentieth century, it was discovered that some filarial nematodes were host to intracellular bacteria [10–12]. Consequently, treatment of filarial worms with tetracycline cleared the bacterial endosymbiont, *Wolbachia pipientis* [13–15], and resulted in fertility defects and the consequent demise of these worms [16–18]. Additional studies showed that *B. malayi* (among other nematodes in the family Onchocercidae) absolutely requires this obligate intracellular bacterium for long-term survival and reproduction [19, 20]. Therefore, identifying the mechanisms essential for the intracellular survival and proliferation *Wolbachia* within *B. malayi* should unveil new molecular targets for treatment of some filarial diseases.

*Wolbachia* has the genetic components to build a functional type IV secretion system (T4SS) [21–24], which is known to be transcribed and translated by *Wolbachia* in vivo [22, 25]. Importantly, reconstitution of the *Wolbachia* T4SS (*w*T4SS) secretion system coupling proteins into an *E. coli*-derived and expressed T4SS apparatus was shown to be capable of translocating a number of *Wolbachia* candidate T4SS effector protein substrates [26], thus strongly implying the functionality and importance of the *w*T4SS during the *Wolbachia*:host endosymbiosis. Although a few *Wolbachia* surface proteins (*w*SPs) have been predicted to be involved in interactions with nematode host proteins [27, 28], the molecular and biochemical characterization of *Wolbachia* Type IV-secreted proteins remains lacking.

Recently, our laboratory screened a number of candidate Type IV-secreted "effector" proteins from the *Wolbachia* endosymbiont of *B. malayi* (*w*Bm) for biological activity in a well-established surrogate model of the eukaryotic cell, *Saccharomyces cerevisiae* [29]. High-level

expression of these candidate effector proteins in yeast revealed several that induced general growth defects and disruptions of protein and membrane trafficking, providing initial clues to their activities in a eukaryotic host cell. One such gene, *w*Bm0076, strongly inhibited yeast growth and induced the formation of aberrant cortical actin patches in the cell upon expression [29]. The Wbm0076 protein is predicted to be a member of Wiskott-Aldrich syndrome proteins (WASp), which regulate conserved core actin polymerization machineries in eukaryotes through the recruitment and activation of the Arp2/3 protein complex to pre-existing actin filaments. There, Arp2/3p nucleates actin monomers (G-actin), thus initiating the polymerization of filamentous actin (F-actin) at a 70˚ angle from the mother filament [30–32]. The formation of these branched actin structures is critical for central cellular pathways, such as motility and endo/phagocytosis [33–35]. As the proper spatiotemporal regulation of actin filamentation by regulatory proteins is often crucial for cellular functions, we hypothesized that Wbm0076 functions as a WAS(p) family protein to directly modulate the actin dynamics in host cells to support *w*Bm colonization and survival in vivo.

In this work, we now show that Wbm0076 behaves as an authentic WASp-family protein in vivo. Overproduction of this protein in yeast strongly inhibits endocytosis by preventing the interaction of branched actin patches with endocytic protein machineries. Structure/function analyses show that mutations introduced into conserved actin-binding or Arp2/3-binding domains of Wbm0076 reduces the overall toxicity of Wbm0076 in yeast. Taken together, these data provide additional molecular evidence that *Wolbachia* produces and secretes proteins required to modulate the actin cytoskeleton of its invertebrate hosts, which may be important for the previously proposed ability of *Wolbachia* to mobilize through host cells via cell-to-cell transmission pathways [36, 37], or for the proper partitioning of *Wolbachia* into *Drosophila* embryos [38, 39]. We expect that this work will pave the way for further studies in *B. malayi* to promote both the understanding of the molecular basis of the *Wolbachia*:host endosymbiosis, and future discoveries of potential inhibitors of the *Wolbachia-B. malayi* relationship.

## Results

### Wbm0076 inhibits yeast endocytosis by decoupling actin patches from sites of endocytosis

To further investigate the impact of *w*Bm0076 expression on yeast actin dynamics and growth toxicity, we considered the possibility that Wbm0076 may be inhibiting clathrin-mediated endocytosis in yeast by aberrantly recruiting and siphoning actin monomers and other endocytosis related actin-patch proteins (eRAPs) from conventional endocytic sites. As increased cell size is a typical phenotype of cells unable to endocytose plasma membrane to balance the delivery of lipids to the cell membrane [40], a lack of endocytosis in *w*Bm0076-expressing cells would explain the enlarged cell phenotype in the presence of Wbm0076 previously observed by our laboratory [29].

The order in which specific endocytic coat proteins and actin patch components arrive to the sites of endocytosis in yeast is relatively well known and can be used to visualize the real-time formation of endocytic vesicles [41] (Fig 1). To determine the effects of Wbm0076 on actin-mediated endocytosis, we expressed *w*Bm0076 in yeast strains harboring GFP fusions marking three different timepoints of actin-mediated endocytosis: Ede1-GFP (early scaffolding protein [42]), Sla1p-GFP (late vesicle coat protein interacting with the actin cytoskeleton [43]), and Sac6-GFP (actin bundling protein [44]). Abp1p binds actin filaments and interacts with several eRAPs at the actin patch, and is thought to a primary regulator of both those eRAPS and local actin dynamics [45]. Therefore, we studied the localization of these eRAPs in comparison to Abp1p-positive cortical actin patches in the presence or absence of Wbm0076, as

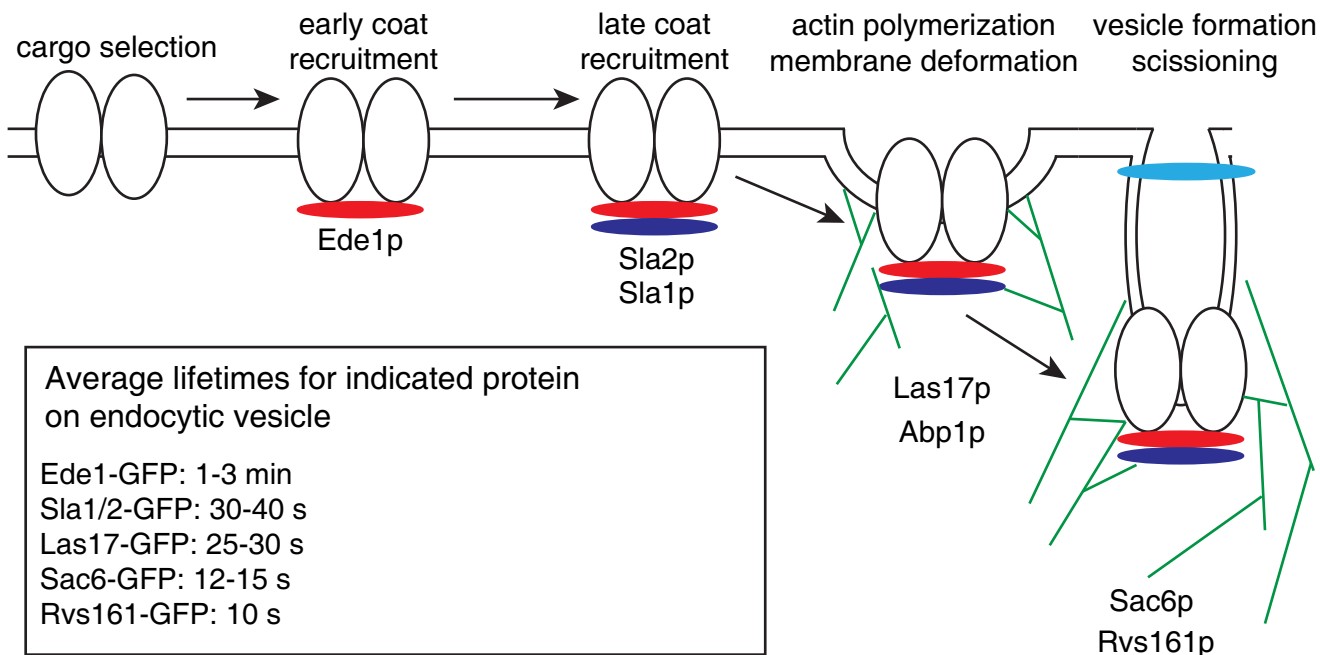

**Fig 1. Highly simplified model of *Sce* endocytic vesicle formation.** Cargo selection recruits early coat proteins (red, Ede1p) and clathrin to endocytic sites. Additional coat proteins are recruited (dark blue, Sla1/2p), then actin and actin polymerization machinery (Las17p, Abp1p). Additional actin polymerization in catalyzed by Sac6p, and vesicles are scissioned via Rvs161p activity. Residence times listed for each protein are based on [41].

the spatial dynamics between these proteins during endocytosis is known [41]. Specifically, the early scaffold membrane protein, Ede1p, does not colocalize with Abp1p, a protein that arrives later in clathrin-mediated endocytosis. Sla1p, on the other hand, is known to interact with eRAPs that connect the actin patch machinery to the budding endocytic vesicle on the yeast membrane [43]. Consequently, Sla1p transiently colocalizes with Abp1p as endocytic vesicles mature and interact with polymerizing branched actin patches [41]. Finally, Sac6p is directly associated with both the actin patch and endocytic vesicle and is therefore expected to colocalize with Abp1p late in endocytosis.

To localize these endocytic markers in the context of branched actin and *w*Bm0076 expression, the galactose-inducible vectors pYES2/NT A control or pYES2/NT A-*w*Bm0076 were introduced into yeast strains constitutively expressing Abp1-mCherry and GFP fusions of Ede1p, Sla1p, or Sac6p. Expression of *w*Bm0076 was driven by β-estradiol-induced chimeric *GAL4* (GAL4.ER.VP16) activation of the *GAL1/10* promoters [46, 47]. We observed that Ede1-GFP is clearly localized at plasma membrane sites that do not colocalize with Abp1-mCherry-positive cortical actin patches, as expected (Fig 2A). Moreover, expression of *w*Bm0076 in this strain did not alter the localization of either Ede1-GFP or Abp1-mCherry, despite the drastic increase in total Abp1-mCherry punctae known to be caused by the expression of *w*Bm0076 in yeast [29] (Fig 2A). In vector control conditions, Sla1-GFP localized predominantly at the plasma membrane and partially colocalized with Abp1-mCherry-positive cortical actin patches, as expected (Fig 2A), while the actin-bundling protein, Sac6p, colocalized with Abp1 nearly perfectly (Fig 2A). After a 6 h β-estradiol induction of *w*Bm0076-harboring strains, Sac6-GFP and Abp1-mCherry are generally colocalized, although the number of Sac6p-GFP colocalized punctae are much more numerous when compared to control conditions, suggesting an increase of Sac6p/Abp1p-containing branched actin patches (Fig 2A). While expression of *w*Bm0076 does not drastically alter the colocalization of Abp1-mCherry

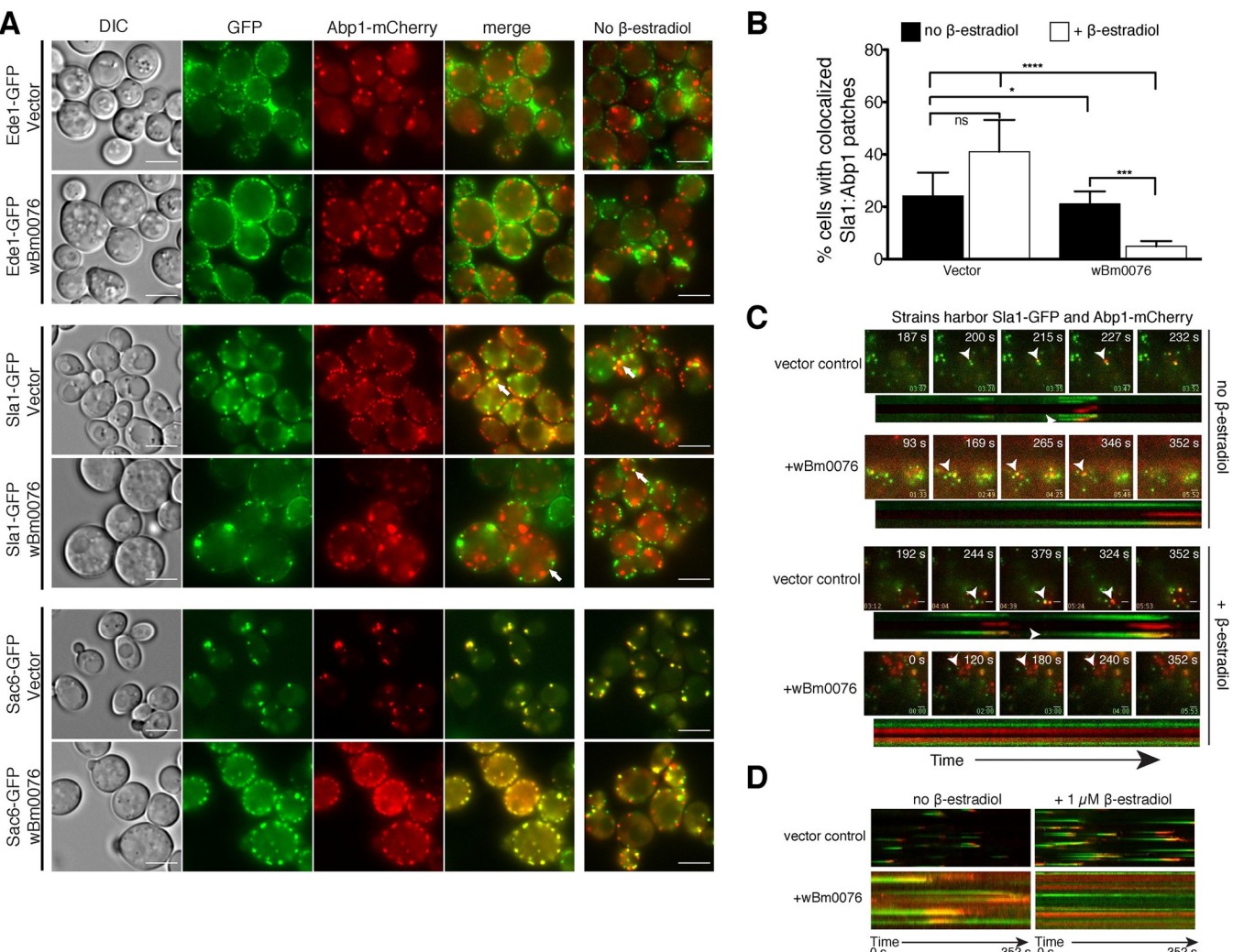

**Fig 2. Wbm0076 disrupts late endocytic vesicle kinetics, but not site selection.** Yeast strains harboring indicated GFP-tagged endocytic vesicle components, Abp1p-mCherry, and either pYES2/NT A (vector) or *w*Bm0076 expression vector were induced for 5 h with 1 μM β-estradiol. Cells were harvested, washed, and **(A)** visualized via epifluorescent microscopy. Right panels are corresponding strains without β-estradiol induction. White arrows point to coupled (vector) or uncoupled (wBm0076) phenotype. **(B)** The percentage of cells showing colocalized patches in Sla1GFP + Abp1RFP strains with vector or *w*Bm0076 expression in **(A)** were calculated (*n* ≥ 100 cells, in triplicate). **(C)** Yeast strains harboring Sla1-GFP and Abp1-mCherry and either a vector control or *w*Bm0076 were induced for 5h in 1μM β-estradiol and visualized via two-color TIRF microscopy over 352 s (3 frames/s). Arrows point to actin patch analyzed by corresponding kymograph. **(D)** Kymograms describing protein kinetics of all patches visualized in a single cell. Bar = 5 μ.

and either Sac6-GFP or Ede1-GFP, we noticed a striking separation of Sla1-GFP and Abp1-mCherry in the presence of Wbm0076 (Fig 2A and 2B). Taken together, these results suggest that *w*Bm0076 expression disrupts the interactions of the late endocytic vesicle coat proteins with the actin cytoskeleton.

To observe the effects of *w*Bm0076 expression on clathrin-mediated endocytosis in real time, we chose to visualize endocytic vesicle formation and internalization using Total Internal Reflection Fluorescence (TIRF) microscopy. TIRF microscopy has long been used to study the kinetics and regulation of yeast endocytosis in detail [48–50]. Using this technique, we can observe endocytic vesicles form at the plasma membrane as fluorescence increases. As endocytic components track back into the cytoplasm to enter endosomal trafficking pathways, however, fluorescence is lost. Because we observed a striking de-localization of Abp1p and Sla1p in

the presence of Wbm0076, we visualized endocytosis in yeast strains harboring both Sla1-GFP (late coat) and Abp1-mCherry (branched actin polymerization). Normal dynamics of endocytic vesicle formation was observed under control conditions, with the initial formation of Sla1-GFP patches (green) at the plasma membrane, Abp1-mCherry colocalization with Sla1p after several seconds (yellow), followed by Sla1p disappearance and then Abp1p "leaving" by being drawn into the cytoplasm (Fig 2C, S1 Movie). This was observed both over the course of a single patch lifetime (Fig 2C, white arrowheads), as well as the lifetime of all patches observed in a single cell over ~ 6 min (Fig 2D). We noted that the addition of β-estradiol alone did appear to slightly increase the membrane residence times of all endocytic vesicles under vector control conditions (Fig 2D, longer tracks; S2 Movie), but given that endocytosis dynamics continued to be otherwise normal overall, we did not investigate this further. In strains harboring a *w*Bm0076 vector, we noted that residence times of both Sla1p and Abp1p were increased under non-induction conditions for *w*Bm0076 (Fig 2B and 2C, S3 Movie), however, patch organization and endocytosis dynamics appeared to follow the proper sequence of events under these conditions overall. In the presence of the 1 μM β-estradiol inducer, however, both Sla1-GFP and Abp1-mCherry proteins were essentially static over the reported time frame (Fig 2C and 2D, S4 Movie). Furthermore, Sla1p and Abp1p were never found to colocalize under these conditions, in support of the strong Sla1p/Abp1p delocalization previously observed in Fig 2A. Taken together, these results show that Wbm0076 strongly inhibits yeast endocytosis by not only inducing the aberrant formation of Abp1p-positive branched actin punctae, but also by preventing the engagement of endocytic membrane components with the branched actin patches required for endocytic uptake in yeast.

## Mutation of transmembrane and VCA subdomains reduces Wbm0076 toxicity in vivo

WAS (p) family proteins are powerful actin regulators involved in the recruitment of the Arp2/3 protein complex along with actin monomers, to promote enhanced actin polymerization [30–32]. Their function is largely directed by a conserved "VCA" domain, where the verprolin/WH2 (V) domain sequesters and binds actin monomers [51], the α-helical central region (C) helps direct the Arp2/3p complex binding to actin monomers [52], and the acidic (A) domain recruits and activates the Arp2/3p complex [30, 53]. WAS(p) proteins are also recruited to the site of actin polymerization and activated via their interactions with WASP-interacting proteins (WIPs), PI(4,5)P$_2$, and Cdc42 at the plasma membrane [54, 55]. While Wbm0076 contains the conserved VCA subdomains found in WAS(p)-family proteins, it also contains a putative N-terminal transmembrane domain which may direct this protein to membranes without a corresponding interaction with a host WIP protein [29, 55].

To assay the importance of the Wbm0076 transmembrane and VCA sub-domains for proper localization and activity of Wbm0076 in vivo, we individually substituted alanine for three highly conserved residues in each of the subdomains (Fig 3, W280A (A domain), R258A (C domain), R209A (V domain)) ([52, 53, 56]). Additionally, to test the requirement of the putative transmembrane helix in vivo, the *w*Bm0076 open reading frame was cloned into the expression vector without the first 61 amino acids (62–392). Expression of all mutant Wbm0076 proteins were confirmed via immunoblot (S1 Fig). We then tested the ability of each of these proteins to inhibit yeast growth, disrupt actin dynamics in vivo, and assayed their localization compared to Abp140-GFP, an actin filament binding protein which binds both actin cables and branched actin patches [48].

As seen previously, expression of *w*Bm0076 is toxic to yeast, and Wbm0076-mRuby2 localizes to a large number of punctae containing Abp140-GFP at the yeast cortex [29]. Removal of

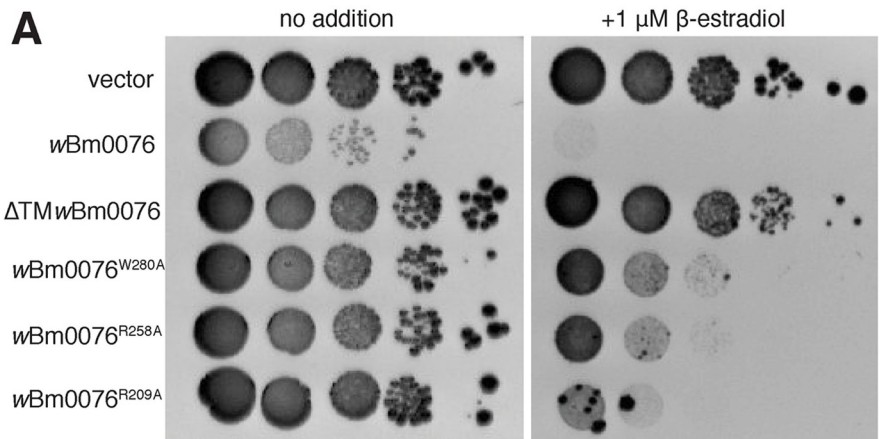

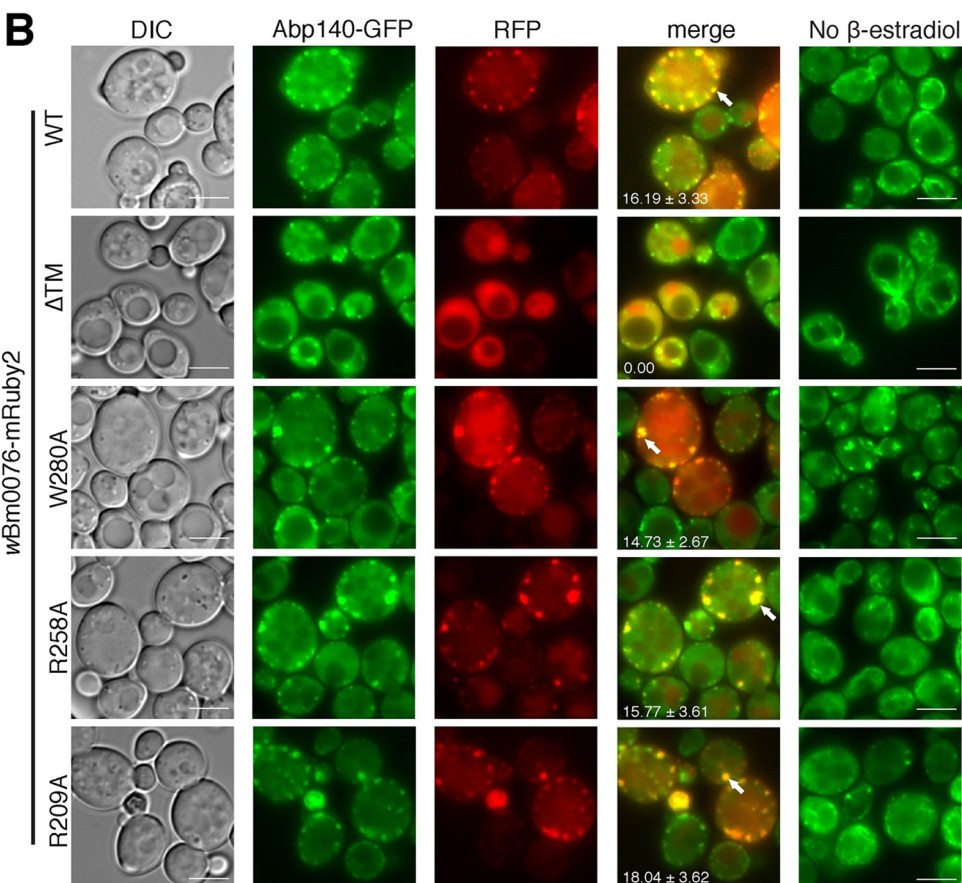

**Fig 3. Wbm0076 domain mutations modify toxic activity.** Yeast strains modified with GEV for β-estradiol-dependent induction of *GAL* promoters (Methods), expressing Abp140GFP and harboring a pYES2/NT A control plasmid or a pYES2/NT A plasmid containing *w*Bm0076 or mutant derivative were grown overnight in CSM medium lacking uracil. **(A)** Cultures were diluted to an $OD_{600}$ = 1.0 in sterile 0.9% NaCl, then spotted in 10-fold dilutions on plates containing or lacking 1 μM β-estradiol. **(B)** Cells were subcultured to fresh CSM-ura containing or lacking 1 μM β-estradiol. After 5 h outgrowth at 30˚C, cells were harvested and visualized. Bar = 5 μ. Number of colocalized actin patches (white arrows) and standard deviation from norm per cell is noted and determined from three independent experiments: $n \geq 100$ cells per experiment. P > 0.05 (n.s.).

the predicted N-terminal transmembrane helix (ΔTM*w*Bm0076), however, completely abrogates the toxicity of Wbm0076 and causes the cytoplasmic accumulation of Wbm0076-mRuby2 (Fig 3A and 3B), suggesting that membrane localization is critical for Wbm0076 activity in eukaryotes *in vivo*. Furthermore, expression of Wbm0076$^{W280A}$, Wbm0076$^{R258A}$, and Wbm0076$^{R209A}$ mutant proteins show reduced toxicity *in vivo* when compared to the wild type (Fig 3A), suggesting the presence of functional 'WH2', 'central', and 'acidic' (VCA) domains in the protein. Wbm0076$^{W280A}$, Wbm0076$^{R258A}$, and Wbm0076$^{R209A}$-mRuby2 continue to show a punctate pattern of recruitment to the cell cortex (Fig 3B), showing that the reduction in toxicity from these mutants is likely due to defects in activation of the Arp2/3p complex by these mutant proteins, although we continue to observe increases in actin patch formation at the plasma membrane in the presence of these mutant proteins as seen by both Abp140-GFP accumulation (Fig 3B) and by staining the entirety of the yeast actin network with rhodamine phalloidin (S2 Fig). We noted that the mutation of the acidic domain had the strongest effect on Wbm0076 toxicity, followed by mutation at the central domain and WH2 domain, respectively. These results support the requirements of membrane localization and functional WAS(p)-family VCA subdomains for Wbm0076 activity in vivo.

## N-terminal membrane association re-establishes Wbm0076 toxicity in yeast

As it is known that WAS(p)-family proteins require proper spatiotemporal regulation at the cell membrane for function [49, 50], we wondered if the toxicity of ΔTM*w*Bm0076 could be restored by replacing the transmembrane domain of Wbm0076 with a membrane-targeting domain. The C2 domain of the lactadherin protein (LactC2) is a phosphatidylserine binding domain, a phospholipid found primarily on the cytosolic leaflet of the yeast plasma membrane [57]. Therefore, we created a chimeric Wbm00076 protein that contains LactC2 in place of the N-terminal transmembrane domain (GFPLactC2-ΔTM*w*Bm0076). To ensure that LactC2 did not colocalize with actin patches at the cell cortex under *w*Bm0076 expression, both GFPLactC2 and *w*Bm0076 were expressed in an Abp1-RFP yeast background; GFPLactC2 alone was not toxic upon expression and did not colocalize with Wbm0076-localized actin patches (Figs 4A and S3). As expected, expression of GFP-*w*Bm0076 remained toxic in the Abp1RFP background strain (Fig 4A), but GFPΔTM*w*Bm0076 was much less toxic (Fig 4A), despite similar levels of protein expression (S4 Fig). Strikingly, the expression of the GFPLactC2-ΔTMWbm0076 protein restored toxicity of the truncated ΔTMWbm0076 protein (Figs 3 and 4A). Furthermore, GFPLactC2-ΔTMWbm0076 was found to colocalize with Abp1RFP at the cell cortex in abnormally large actin patches (Fig 4B, arrowhead), showing that localization to membranes is critical for both the toxicity and presumed Arp2/3-activating activity of Wbm0076 in vivo.

## Wbm0076 co-precipitates with Abp1p

Abp1p is a highly-conserved nucleation promotion factor (NPF) that binds actin filaments with its N-terminal actin depolymerizing factor homology (ADFH) domain and recruits the Arp2/3 protein complex to these filaments with its acidic domains, thus promoting the actin nucleation activity of Arp2/3 [45, 58]. Abp1p is also thought to organize the cortical actin patch via interactions with a number of eRAPs using its ADFH and polyproline-binding SH3 domain [45, 59, 60]. Therefore, Abp1p activity is critical for regulating actin dynamics and endocytosis and plays an important role in mammalian cortical actin dynamics [61]. In our previous study, we noted that *abp1Δ* yeast strains reduced Wbm0076 toxicity in yeast [29]. Moreover, in various microscopy assays, Abp1p-RFP/mCherry localizes with Wbm0076 at

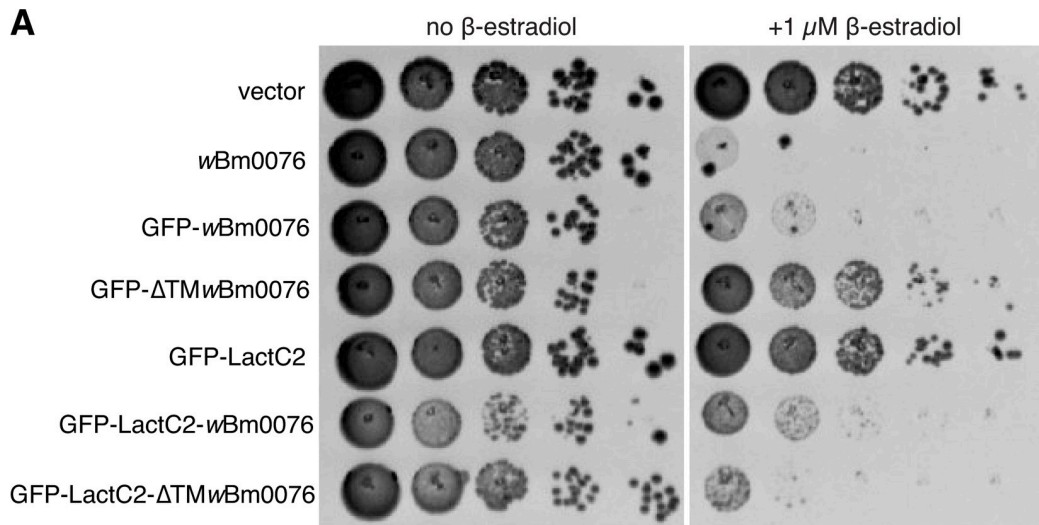

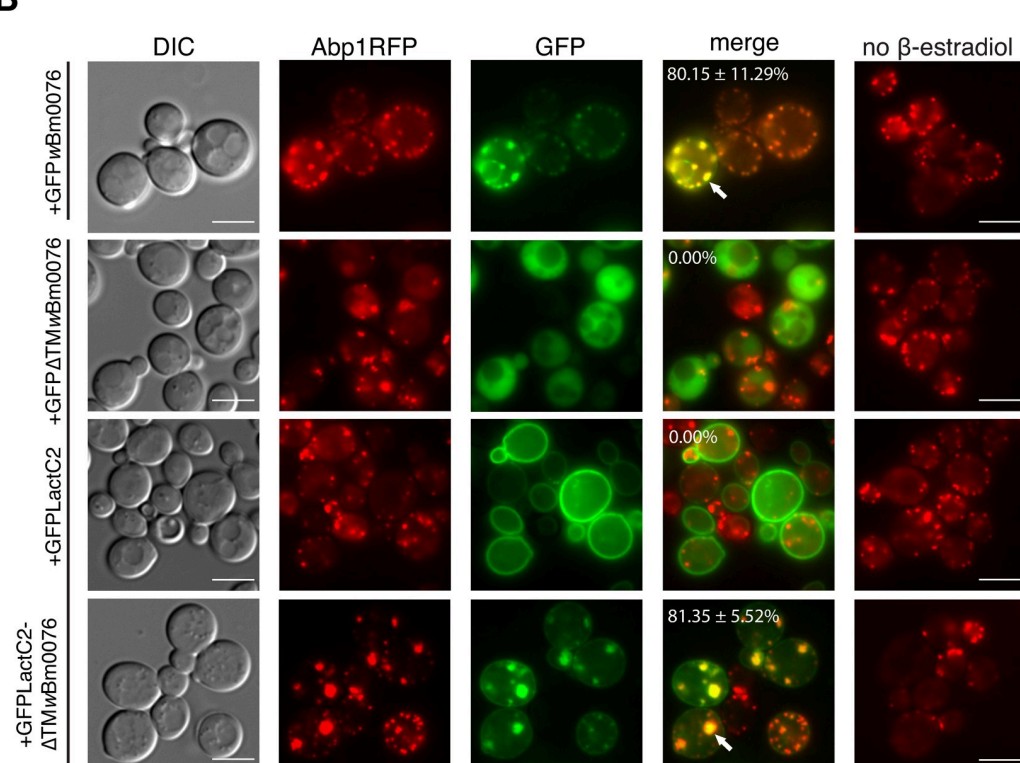

**Fig 4. Wbm0076 activity requires membrane association.** Yeast strains modified with GEV for β-estradiol-dependent induction of *GAL* promoters expressing Abp1RFP and harboring a pYES2/NT A control plasmid or a pYES2/NT A plasmid containing the indicated construct were grown overnight in CSM medium lacking uracil. **(A)** Cultures were diluted to an $OD_{600}$ = 1.0 in sterile 0.9% NaCl, then spotted in 10-fold dilutions on plates containing or lacking 1 μM β-estradiol. **(B)** Cells were subcultured to fresh CSM-ura containing or lacking 1 μM β-estradiol. After 5 h outgrowth at 30˚C, cells were harvested and visualized. Number of cells with colocalized actin patches (white arrows) and the standard deviation from the norm is noted and determined from three independent experiments: $n \geq 100$ cells per experiment, bar = 5 μ.

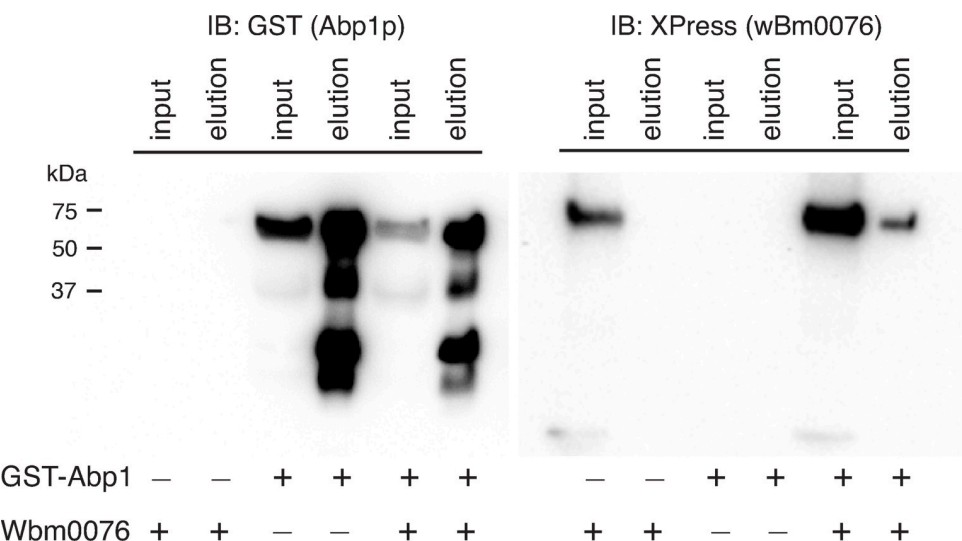

**Fig 5. Wbm0076 coprecipitates with GST-Abp1p.** Protein extracts from indicated strains were generated and incubated with glutathione beads as in Methods. Equal fractions of input and elution for each condition were separated via SDS-PAGE and immunoblotted with the indicated antibody (Anti-Xpress: Wbm0076; anti-GST, GST-Abp1). Due to the similarity in the sizes of GST-Abp1 and Wbm0076, two identical gels and membranes were probed with different primary antibodies.

aberrant actin patches (Figs 3 and 4). As Wbm0076 contains various polyproline regions [29], we wondered whether Wbm0076 physically interacts with Abp1p in vivo.

To assess the possibility of this interaction, protein extracts from strains expressing only GST-Abp1p, only Wbm0076(containing an N-terminal Xpress epitope), or both proteins were incubated with glutathione beads, washed, and bound proteins eluted. GST-Abp1 was effectively pulled down from cell lysates under these conditions, as expected (Fig 5, left panel), while Wbm0076 alone did not interact with the glutathione beads (Fig 5, right panel). From cells harboring both pWbm0076 and pGST-Abp1p, however, Wbm0076 was clearly detected in the GST-Abp1 pulldown (Fig 5, right panel), suggesting either an indirect or direct interaction between these proteins in vivo.

## Discussion

*Saccharomyces cerevisiae* remains a powerful model of general eukaryotic biology, especially in the fields of protein and membrane trafficking (reviewed in [62]), endolysosomal membrane dynamics (reviewed in [63]), and cytoskeletal dynamics (reviewed in [64]). As these pathways are critical across eukaryotic cellular physiology, including nematodes, the structural and regulatory proteins of these pathways are typically conserved. Due to this fact, the yeast cell model has long been used to characterize secreted effector proteins from a number of pathogenic intracellular bacteria, as modulation of these eukaryotic pathways by these bacterial proteins can result in yeast growth defects [65–67]. Importantly, while some of these bacterial effectors are lethal to yeast upon expression, results obtained from these screens do not necessarily suggest that the physiological activity of the bacterial protein is meant to intoxicate host cells, but rather help provide crucial information regarding the potential eukaryotic pathways which may be modulated by the activities of these proteins. Accordingly, we have recently used this system to identify and begin the characterization of 47 candidate T4SS effector proteins from the uncultivable *Wolbachia* endosymbiont of *B. malayi*. Of those 47, Wbm0076 demonstrated a unique ability to strongly disrupt normal yeast actin dynamics, resulting in cell lysis and

death presumably through the reduction or elimination of cellular F-actin structures ([29], S2 Fig); this phenotype is similar to what has been previously observed in vegetatively-growing yeast cells which completely lack F-actin structures [68]. With this work, we now provide additional evidence that Wbm0076 functions as a WAS-like protein in vivo to directly modulate conserved eukaryotic actin dynamics.

In this study, we observed that Wbm0076 is localized to actin patches at the cortex of the yeast cell, where its overall cellular toxicity is dependent on its conserved VCA subdomains, presumably due to its ability to drive actin polymerization (Fig 3) [29]. Endocytic vesicle invagination is likely inhibited in strains expressing *w*Bm0076 due to the improper engagement of branched actin with endocytic vesicles after they have begun to form, as shown by Sla1p-GFP: Abp1p-mCherry mislocalization (Fig 4). Combined with the increase in actin patch numbers upon *w*Bm0076 expression [29], these results suggest that perhaps Wbm0076 may be initiating the formation of *de novo* actin patches that are uncoupled from endocytic sites; a phenotype that has been previously seen in *sla2Δ* yeast strains, where the actin patch is weakly linked to membrane invagination, and actin comet tail-like structures, detached from the plasma membrane, are formed [69]. We also find that the transmembrane domain of Wbm0076 is required for membrane localization and subsequent activity in vivo (Fig 2A and 2B). Removal of the predicted N-terminal transmembrane helices completely abrogates the toxicity of Wbm0076 and causes the cytoplasmic accumulation of Wbm0076 (Fig 2A and 2B). However, complementation of the transmembrane helices with the phosphatidylserine probe and membrane targeting domain, lactadherin C2, restores Wbm0076 toxicity in vivo (Fig 3A and 3B), showing that membrane localization is critical for Wbm0076 activity in vivo, presumably by placing Wbm0076 in proximity to the Arp2/3 complex and other regulatory proteins.

Interestingly, eukaryotic WAS(p)-family proteins are not known to contain transmembrane domains, but rather rely on interactions with other membrane-binding proteins to provide both membrane recruitment and subsequent activation of the WAS(p) protein [70]. For example, WASP (yeast Las17p) and N-WASP proteins are different from other WAS(p) family proteins in that they contain an N-terminal WIP (yeast Vrp1p) binding domain (WH1) [55, 71, 72]. By removing the requirement of interacting with other membrane-binding proteins for membrane recruitment, Wbm0076 (and other *Wolbachia* orthologs (Fig 6)) can be immediately placed into membranes post-secretion from the bacterium and initiate cytoskeletal rearrangements important for the intracellular lifestyle of *Wolbachia*.

The use of WAS(p)-family proteins by other intracellular bacteria to manipulate host actin dynamics is a well-known phenomenon. Important WAS(p)-family protein members secreted by bacteria include RickA and ActA of *Rickettsia conorii* and *L. monocytogenes*, respectively [73, 74]. These surface-exposed proteins recruit Arp2/3 to the bacterial cell wall and force the polymerization of actin to create actin "comet tails," a branched network of short actin filaments that are continuously severed into monomers and repolymerized to provide the force to help the bacteria move through their host cells and support the invasion of the bacterium into neighboring cells [74–77]. In the many ultrastructural studies of the nematode:*Wolbachia* relationship, however, actin comet tails surrounding intracellular *Wolbachia* bacteria have never been observed [78–80] However, other studies have suggested that *Wolbachia* utilizes host cortical actin machinery to mobilize across cells during nematode development, as well as requiring normal host actin dynamics to partition correctly into developing embryos [36] and clathrin-mediated endocytosis to promote endocytosis into *Drosophila* cells [37]. Therefore, we find it likely that *Wolbachia* utilize secreted Wbm0076 orthologs to subvert local actin dynamics to regulate the movement of bacteria through and into host cells.

In a previous study performed in our lab, we noted that Wbm0076 toxicity was reduced in *abp1Δ* yeast strains [29], suggesting that Abp1p protein activity may be important for the in

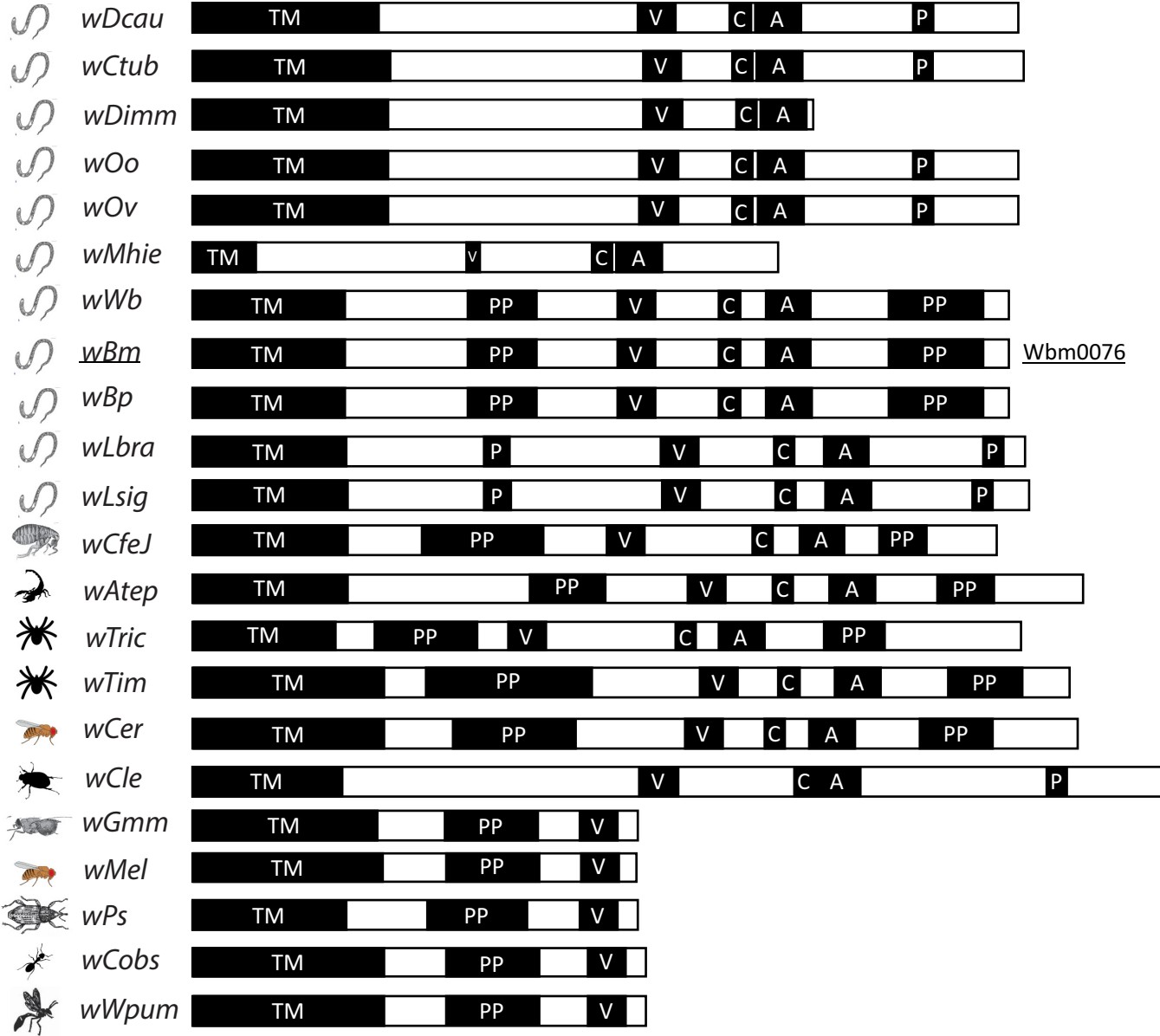

**Fig 6. Wbm0076 is conserved among *Wolbachia* endosymbionts.** Protein sequences from the *Wolbachia* genus homologous to Wbm0076 (only 22 of 31 shown here) were identified via the Blastp suite [91] and are indicated with abbreviations of the species of origin. Sequence alignment of conserved domains are highlighted with a black box. TM = transmembrane; V = verprolin/WH2 domain; C = central domain; A = acidic domain; P/PP = polyproline motif. Species abbreviations and NCBI accession numbers for all 31 identified homologues are available in S5 Fig.

vivo activity of Wbm0076. Interestingly, in metazoans, Abp1p homologs have been shown to interact with WASP/N-WASP, WIPs, and the GTPase dynamin to link the actin dynamics machinery to the endocytic machinery and to promote the scission of the mature endocytic vesicle [61, 81–83]. We have confirmed that Wbm0076 co-precipitates with GST-Abp1 in yeast, suggesting that Wbm0076 either directly interacts with conserved Abp1p, or is a part of a larger complex of actin regulatory proteins that contain Abp1p.

This result warrants further study, as in *B. malayi*, *Wolbachia* mobilizes from the hypodermal tissues to the reproductive organs as the microfilariae mature. In nematodes where *Wolbachia* has been previously cleared by antibiotic treatment, aberrant development of the

nematode occurs, resulting in germline apoptosis and embryogenesis defects in the nematode that render *B. malayi* infertile [20, 84]. *Wolbachia* have been observed in the pseudocoelomic space of L4 and adult worms, interacting with the membrane at the distal tip of filarial ovaries [36, 78]. Moreover, it has been found that transcription levels of the *w*Bm0076 gene are highest in the body wall and ovaries of the adult female *B. malayi*, suggesting the effector is involved in either *w*Bm exit from body wall tissues or invasion of ovaries in this maturation step [85]. *B. malayi* contains homologs of WIP (*Bm5420*), Abp1p (*Bm4914*), dynamin (*Bm1908*), and all the subunits of the Arp2/3 complex (BMA-ARX complex). Due to the Wbm0076-Abp1p interaction we observed, we believe that *w*Bm utilizes Wbm0076 as a WASP-like protein to target host cortical actin structures to either promote its endocytic uptake into both gonads and neighboring cells, or to drive the actin polymerization needed to properly segregate *Wolbachia* into embryos during the maternal transmission event. As *w*Bm0076 orthologs exist throughout *Wolbachia* (Fig 6), we feel that this is a common theme across *Wolbachia*:host symbioses.

It is important to note that the activity of Wbm0076 observed in our yeast model may not accurately represent the authentic activity of this protein in the nematode host. Higher eukaryotic cells, such as those found in *B. malayi*, express additional regulators of the actin cytoskeleton which are missing in yeast, such as SCAR/WAVE proteins [86]. Therefore, the ability of Wbm0076 to regulate actin dynamics in yeast may not exactly mirror its activity in the host due to other host-produced regulators. Furthermore, even though we observed Wbm0076 was a potent inhibitor of yeast endocytosis and induced cell death, expression levels of *w*Bm0076 in this model cell would be vastly greater than what *Wolbachia* could deliver to a host cell during its endosymbiosis, and proper gene dosage may also be an important mechanism to regulate Wbm0076 activity. Wbm0076 would also be utilized as part of a cocktail of effectors secreted into the host by the bacterium, some of which may also provide the proper spatiotemporal regulation of Wbm0076 lacking in our study. Despite these limitations, however, we show here that Wbm0076 is a *Wolbachia* actin dynamics-modulating protein in a eukaryotic cell and aspects of its activity are likely to be conserved in *Brugia malayi*. Therefore, continued analysis of the Wbm0076 binding targets (both host and bacterial) and determining the authentic localization of Wbm0076 in *B. malayi* will likely identify the molecular requirements for the actin-dependent pathways that support the intracellular persistence of *Wolbachia* in hosts.

## Methods

### Yeast strains and plasmid constructions

For the microscopy in Figs 2 and 3, yeast strains were derivatives of SEY6210a (*MATa ura3-52 leu2-3, 112 his3-Δ100 trp1-Δ901 lys2-801 suc2-Δ9)*, obtained as a kind gift from Dr. Derek Prosser. For all other studies, yeast strain BY4742 (*MATα his3Δ1 leu2Δ0 lys2Δ0 ura3Δ0*) was used. In order to create yeast strains that activate *GAL1* promoters via the addition of β-estradiol, strains were transformed with linearized pAGL (a gift from Dr. Daniel Gottschling, University of Washington), which introduces the gene encoding for the Gal4-estrogen receptor-VP16 (GEV) chimeric protein into the *leu2*Δ0 locus [46]. S288C yeast strains expressing either Abp1p-RFP or Abp140-3xGFP were a kind gift from Dr. Bruce Goode (Brandeis University).

To create the *w*Bm0076-mRuby2 expressing pYES2NTA plasmid, the yomRuby2 gene was amplified from the plasmid pFA6a-link-yomRuby2-SpHis5 [87] using primers 5'-AGCTTTTCTTATAAAACAATTGATGGTGTCCAAAGGAGAGGAG and 5'- AGGGA-TAGGCTTAGCTGCAATTTACTTATACAATTCATCCATA, containing homology to both the C-terminus of the *w*Bm0076 gene and the pYES2NTA-*w*Bm0076 vector. BY4742-pAGL was co-transformed with pYES2NTA-*w*Bm0076, previously digested with PmeI, and the

mRuby2 amplicon and were plated to CSM-uracil to select for gap-repaired plasmids. Transformants were screened for red fluorescence via microscopy.

All plasmid clones were purified and sequenced for confirmation (Eton Bioscience Inc.)

## Microscopy

β-estradiol responsive yeast strains harboring indicated plasmids were grown to saturation overnight in selective medium at 30˚C, subcultured to fresh media with or without 1 µM β-estradiol and grown for an additional 5 hours. After 5 hours, the entire culture was harvested via centrifugation and cell pellets suspended in 50 µL sterile water. Cell suspensions were mounted to slides pre-treated with a 1:1 mixture of polylysine (10% w/v):concanavalin A (2 mg/ml) solution. Cells were visualized using a Nikon Ti-U fluorescence microscope, and images were processed using the Fiji software package [88, 89].

To measure endocytosis dynamics via TIRF microscopy, β-estradiol responsive yeast strains harboring the indicated fluorescent proteins were cultured overnight in selective media at 30˚C. Saturated cultures were then subcultured to the same media either lacking, or containing, 1 µM β-estradiol to induce *w*Bm0076 expression for 5 h. Cells were mounted on coverslips pretreated with a 1:1 solution of concanavalin A (2 mg/ml) and 0.1% polylysine. For imaging, we used an Eclipse Ti-U microscope (Nikon) equipped with 60× NA1.49 TIRF objective and through-the-objective TIRF illumination provided by a 40-mW 488-nm and a 75-mW 561-nm diode laser (Spectraphysics) as previously described [90]. The excitation lasers were cleaned up with a Nikon GFP/mCherry TIRF filter cube and the emission was separated using an Image Splitting Device (Photometrics DualView2 with filter cube 11-EM). Images were recorded at 10 fps using an iXON3 (Andor) and the NIS-Elements Advanced Research software (Nikon). Frames were assembled into movies and kymograms describing individual patch dynamics were generated using the Fiji distribution of ImageJ (v.1.48s).

## Pulldowns

Yeast strains were cultured overnight in CSM selective media to saturation at 30˚C. Cells were then harvested via centrifugation (3000 x *g*, 5 min, room temperature), resuspended in either YP or CSM-uracil medium containing 2% galactose, and grown for an additional 6 hours at 30˚C. 100 $OD_{600}$ units were harvested from each condition via centrifugation. Cells were then resuspended with ice-cold 1x PBS buffer with 1mM PMSF and protease inhibitor cocktail. Cells were harvested again via centrifugation at 7000rpm for 1min (4˚C). The resultant supernatant was discarded, and the cell pellet was frozen in liquid nitrogen. The cells were thawed and resuspended again in ice-cold 1x PBS buffer (1 mM PMSF, 1x PIC, and 1% (v/v) Triton X-100). Acid-washed glass beads (0.5 µ) were then added to slurry and the cells were mechanically disrupted with a Mini-Beadbeater (Biospec Products) at 4˚C for 20 sec, then placed on ice for 1 min (7 cycles). The lysate was nutated for 1 hour at 4˚C, and insoluble material was cleared by centrifugation (21000 *x g*, 15 min, 4˚C). Cleared lysate was then loaded unto pre-washed Glutathione agarose resin and nutated for 5 hours at 4˚C. Beads were washed 5 times with 1x PBS buffer containing 1mM PMSF, 1x PIC, and 1% Triton X-100. The beads were boiled in 1x SDS loading buffer to release bound protein before western blotting.

## Statistical analysis

Statistical analysis was performed within the Prism software package (GraphPad Software, v. 6.0b). Column statistics were performed via a 1-way ANOVA Repeated Measures test and Holm-Bonferroni post-test. Where noted in figures, ns = P > 0.05 (not significant); (*) = P ≤ 0.05; (**) = P ≤ 0.01; (****) = P ≤ 0.0001.

## Supporting information

**S1 Fig. Western blot of *w*Bm0076 mutant expression.** Yeast strains harboring a pYES2/NT A control plasmid, or a pYES2/NT A plasmid cloned with one of the following: *w*Bm0076, ΔTM*w*Bm0076, *w*Bm0076 (W280A), *w*Bm0076 (R258A), *w*Bm0076 (R209A) were grown overnight in CSM medium lacking uracil. Cells were subcultured to fresh CSM-Ura containing or lacking 1 μM β-estradiol. After 5 h outgrowth at 30˚C, cells were lysed, boiled in SDS-PAGE loading buffer, and loaded for western blot analysis. Anti-Xpress antibodies were used to probe the presence of each protein.
(TIF)

**S2 Fig. *w*Bm0076 expression disrupts the yeast actin network.** Yeast strains harboring a pYES2/NT A control plasmid, or a pYES2/NT A plasmid cloned with one of the following: *w*Bm0076, *w*Bm0076 (W280A), *w*Bm0076 (R258A), were grown overnight in CSM medium lacking uracil. Cells were subcultured to fresh CSM-Ura and outgrown for 2h at 30˚C with shaking. The actin cytoskeleton was stained with rhodamine phalloidin (S1 Methods) and imaged; bar = 5 μ.
(TIF)

**S3 Fig. LactC2-GFP does not colocalize with *w*Bm0076.** Yeast strains modified with GEV for β-estradiol-dependent induction of *GAL* promoters (Methods), expressing both Abp1RFP and LactC2-GFP, and harboring either a pYES2/NT A control plasmid or a pYES2/NT A plasmid cloned with *w*Bm0076 were grown overnight in CSM medium lacking uracil. Cells were subcultured to fresh CSM-Ura containing or lacking 1 μM β-estradiol. After 5 h outgrowth at 30˚C, cells were harvested and visualized. Bar = 5 μ.
(TIF)

**S4 Fig. Western blot of LactC2-*w*Bm0076 fusion expression.** Yeast strains expressing Abp1-RFP and harboring either a pYES2/NT A control plasmid, or a pYES2/NT A plasmid cloned with one of the following: *w*Bm0076, GFP-*w*Bm0076, GFP-ΔTM*w*Bm0076, GFP-LactC2, GFP-LactC2-*w*Bm0076, or GFPLactC2-ΔTM*w*Bm0076, were grown overnight in CSM medium lacking uracil. Cells were subcultured to fresh CSM-Ura containing or lacking 1 μM β-estradiol. After 5 h outgrowth at 30˚C, cells were lysed, boiled in SDS-PAGE loading buffer, and separated via SDS-PAGE. After protein transfer, the membrane was bisected at ~40 kDa and the corresponding antibodies were used to detect *w*Bm0076 derivatives (anti-Xpress) or the Sec17p loading control (anti-Sec17p).
(TIF)

**S5 Fig. Alignment of Wbm0076 homologs.** Clustal Omega multiple sequence alignment [92] of *Wolbachia* proteins homologous to Wbm0076 (WP_011256278). The NCBI accession numbers are as follows: *wDcau* (*Dipetalonema caudispina*)–WP_175809232.1, *wCtub* (*Cruorifilaria tuberocauda*)–WP_175813987.1, *wDimm* (*Dirofilaria immitis*)–WP_175818612.1, *wOo* (*Onchocerca ochengi*)–WP_014868845.1, *wOv* (*Onchocerca volvulus*)–WP_025264345.1, *wCle* (*Cimex lectularius*)–WP_052463318.1, *wMhie* (*Madathamugadia hiepei*)–WP_175937123.1, *wWb* (*Wuchereria bancrofti*)–WP_088415029.1, *wBp* (*Brugia pahangi*)–WP_167896226.1, *wLbra* (*Litomosoides brasiliensis*)–WP_175939083.1, *wLsig* (*Litomosoides sigmodontis*)–WP_175837390.1, *wCfeJ* (*Ctenocephalides felis*)–WP_174855505.1, *wPs* (*Pissodes strobi*)–MBV2146497.1, *wAtep* (*Atemnus politus*)–WP_174132891.1, *wTric* (*Trichonephila clavata*)–GFQ77688.1, *wTim* (*Trichonelphila inaurata madagascariensis*)–GFY79972.1, *wCer* (*Rhagoletis cerasi*)–WP_213863040.1, *wGmm* (*Glossina morsitans morsitans*)–WP_240991797.1, *wVitA* (*Nasonia vitripennis*)–ONI57582.1, *wCobs* (*Cardiocondyla obscurior*)–WP_174515724.1,

*wWpum (Wiebesia pumilae)*–WP_209472625.1, *wInc (Drosophila incompta)*–AOV87985.1, *wMuni (Muscidifurax uniraptor)*–ONI56184.1, *wMel (Drosophila Melagonaster)*–ERN55482.1, *wOgib (Oedothorax gibbosus)*—WP_250295490.1, *wCauA (Carposina sasakii)*—WP_246039501.1, *wKgib (Kradiba gibbosae)*—WP_242049606.1, *wCsol (Ceratosole solmsi)*—WP_244662783.1, *wStriCN (Laodelphax striatellus)*—WP_063631094.1, *wAdent (Apterostigma dentigerum)*—NEV49289.1
(EPS)

**S1 Methods. Phalloidin staining of the yeast actin cytoskeleton.**
(DOCX)

**S1 Movie. Control real-time endocytosis in yeast.** Yeast strain harboring Sla1-GFP and Abp1-mCherry and a vector control was visualized via two-color TIRF microscopy over 352 s (3 frames/s), following details in Methods.
(AVI)

**S2 Movie. Control real-time endocytosis in yeast under induction conditions.** Yeast strain harboring Sla1-GFP and Abp1-mCherry and a vector control was induced for 5h in 1μM β-estradiol, then was visualized via two-color TIRF microscopy over 352 s (3 frames/s), following details in Methods.
(AVI)

**S3 Movie. Real-time endocytosis in yeast under *w*Bm0076 non-induction conditions.** Yeast strain harboring Sla1-GFP and Abp1-mCherry and the inducible *w*Bm0076 vector was visualized via two-color TIRF microscopy over 352 s (3 frames/s), following details in Methods.
(AVI)

**S4 Movie. Real-time endocytosis in yeast under *w*Bm0076 induction conditions.** Yeast strain harboring Sla1-GFP and Abp1-mCherry and the inducible *w*Bm0076 vector was induced for 5h in 1μM β-estradiol, then was visualized via two-color TIRF microscopy over 352 s (3 frames/s) following details in Methods.
(AVI)

## Acknowledgments

The authors would like to thank Drs. Bruce Goode and Derek Prosser for providing essential reagents. pLact-C2-GFP-p416 was a gift from Sergio Grinstein (Addgene plasmid # 22853). The content is solely the responsibility of the authors and does not necessarily represent the official views of the National Institutes of Health.

## Author Contributions

**Conceptualization:** Michael K. Mills, Vincent J. Starai.

**Data curation:** Michael K. Mills.

**Funding acquisition:** Vincent J. Starai.

**Investigation:** Michael K. Mills, Lindsey G. McCabe, Eugenie M. Rodrigue, Vincent J. Starai.

**Methodology:** Michael K. Mills, Lindsey G. McCabe, Eugenie M. Rodrigue, Karl F. Lechtreck, Vincent J. Starai.

**Project administration:** Vincent J. Starai.

**Resources:** Karl F. Lechtreck.

**Supervision:** Karl F. Lechtreck, Vincent J. Starai.

**Writing – original draft:** Michael K. Mills.

**Writing – review & editing:** Michael K. Mills, Vincent J. Starai.

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
