## [Decision Letter · Decision Letter 0]

10 Oct 2022

Dear Dr. Starai,

Thank you very much for submitting your manuscript "Wbm0076, a candidate effector protein of the Wolbachia endosymbiont of Brugia malayi, disrupts eukaryotic actin dynamics" for consideration at PLOS Pathogens. As with all papers reviewed by the journal, your manuscript was reviewed by members of the editorial board and by several independent reviewers. In light of the reviews (below this email), we would like to invite the resubmission of a significantly-revised version that takes into account the reviewers' comments.

If you choose to submit a revised version of the manuscript, please make sure to highlight that the findings in yeast do not necessarily represent the effects this protein may have in Brugia nematodes. I feel this needs to be clearly stated as an important limitation upfront in the discussion. Also, please provide alternative possible explanations for the findings as highlighted by reviewer 2.

We cannot make any decision about publication until we have seen the revised manuscript and your response to the reviewers' comments. Your revised manuscript is also likely to be sent to reviewers for further evaluation.

Sincerely,

Edward Mitre

Associate Editor

PLOS Pathogens

James Collins III

Section Editor

PLOS Pathogens

Kasturi Haldar

Editor-in-Chief

PLOS Pathogens

orcid.org/0000-0001-5065-158X

Michael Malim

Editor-in-Chief

PLOS Pathogens

orcid.org/0000-0002-7699-2064

If you choose to submit a revised version of the manuscript, please make sure to highlight that the findings in yeast do not necessarily represent the effects this protein may have in Brugia nematodes. I feel this needs to be clearly stated as an important limitation upfront in the discussion. Also, please provide alternative possible explanations for the findings as highlighted by reviewer 2.

Reviewer's Responses to Questions

**Part I - Summary**

Reviewer #1: Wolbachia pipientis is an important and ubiquitous symbiont in insects, nematodes, and arthropods more broadly. We know from work in the field that Wolbachia expresses a type IV secretion system and that it likely uses its secreted effectors to modify host biology. Understanding the evolution and function of these effectors is important and here, the authors follow up on their large scale yeast screen identifying candidate effectors in the Wolbachia that infect the nematode Brugia malayi. They show that one protein from the Wolbachia from Brugia (wBm0076) co-localizes with a couple of actin binding proteins (Abp1p and Abp140) and alters vesicle trafficking dynamics in the cell. I was a bit confused by the title as it downplays the vesicle work and the authors don’t ever actually show the actin cytoskeleton of the yeast are altered by the presence of wBm0076 but instead focus on Abp1p as a proxy for actin dynamics. It would’ve been nice to see some basic phalloidin staining. Overall, I think this is a nice study but I have a few suggestions for the authors below. Additionally, there were many typos throughout with regards to figure citations and the use of correct nomenclature for Wolbachia proteins. I suggest they carefully read through the manuscript before resubmission.

Reviewer #2: This study is a continuation of the author's previous work on the interaction of Wolbachia endosymbiont of Brugia malayi proteins with actin and membrane-interacting proteins of yeast, Saccharomyces cerevisiae. The said proteins cannot be easily studied in the native Wolbachia-worm symbiosis, as both organisms are genetically intractable. Previous work has already established the toxicity of the putative Wolbachia effector, Wbm0076, in yeast cells and suggested that the interaction of this heterologously expressed protein with the membrane might lead to cell lysis. This manuscript provides a genetic dissection of the mechanism of toxicity of Wbm0076, and postulates that the phenotype observed in yeast (cell killing) can have a different manifestation in the native host (promoting of Wolbachia endocytic uptake by host cells). Yest genetics experiments are rigorous, they include all required controls.

**Part II – Major Issues: Key Experiments Required for Acceptance**

Reviewer #1: Figure 2: As the authors note in the text (lines 198-200), it seems like the biggest phenotype is the large number of Abp1-mCherry foci per cell - this should be quantified and statistical differences noted.

Line 295 “its ability to produce aberrant branched actin structures is dependent on its conserved VCA subdomains (Fig 2)” - do you mean Figure 3? Quite frankly, I do not think that Figure 3 shows this - there is no actin staining and Abp140-GFP punctae are not quantified. Why use Abp140GFP as a proxy for actin dynamics? Why switch from Abp1p? Why not stain actin?

I love Figure 4 - the complementation of toxicity with LactC2 is striking! I wonder, do the WH2, central, and acidic mutants show different localization than WT with regards to co-localization of Abp1p? Similarly, I wonder if you could quantify Abp1pRFP punctae in this background compared to vector alone (as requested above).

With regards to Figure S3 - the western has two band sizes here - what is that lower band? The TMWbm0076 does indeed look to have less expression when compared to the other columns - where is your loading control? Would be good to add because as it is, it seems like your reduced toxicity could be linked to reduced.

Reviewer #2: The conclusions about Wbm0076 toxicity, its interactions with actin and endocytosis machinery are well validated, both here and in Carpinone et al. 2018.

I have three clarification requests, mostly considering data interpretation and importance of Wbm0076 in the native Brugia-Wolbachia symbiosis:

1. What is the timeline of events here? How long after Wbm0076 interaction with endocytic machinery do yeast cells die?

2. Does interaction of Wbm0076 with membrane causes cell puncture at the site of interaction?

3. The reasoning behind toxicity in yeast turning into encocytosis promoting-phenotype in native Wolbachia hosts is unclear to me. What is the "physiological" reason for yeast cell death here? Do they starve as a result of endocytic pathway distruption? And how do they die, is it cell lysis or, eg. apoptosis?

Also, in the endocytosis-promoting function, Wolbachia would be outside of the host cell; where would the Wbm0076 be? On its surface? Or in the host cell already?

I am asking as the cell killing caused by Wbm0076 can be interpreted differently. As Brugia malayi cannot survive without Wolbachia, the effector studied here could be a Wolbachia life-insurance: killing of Wolbachia cells could release this effector and kill host cells. Thus, under normal conditions Wmb0076 could be strictly intracellular. As this is a much simpler conclusion, and it seems to fit with the data presented here better, I wonder whether it is a possibility in this system.

Another option is consistent with the interpretation provided here (the importance of Wbm0076 in cell-to-cell movement of Wolbachia) but suggests that the Wbm0076 would enable Wolbachia exit from already infected cells rather that its subsequent uptake. Studying the native distribution of Wolbachia in Brugia seems important to discern between these two options. This seems especially important as authors postulate conservation of endocytic machinery between all eukaryotes (hence the heterologous system usage) yet predict a completely different interaction outcomes between the model used here and the native Wolbachia hosts.

**Part III – Minor Issues: Editorial and Data Presentation Modifications**

Reviewer #1: Line 83-85: is it Wolbachia secreted proteins (wSP’s) or Wolbachia surface proteins? Pick one please.

Line 110-112 - I would say this work is excellent genetic evidence that Wolbachia encodes a protein that modulates the actin cytoskeleton. The production and secretion and the “critical” - not so much supported by these data. You can speculate only.

Line 221 - you mean Fig S3 here - please check throughout the manuscript to ensure you refer to the correct figures and data files.

I suggest adding a citation to Figure 6 to line 319 as well.

Line 362 - should be wBm0076 (not Wbm0076) - check throughout - saw this also on line 293.

Line 335 - actin has also been shown to be important for Wolbachia’s maternal transmission in fruit flies (see Sheehan et al., 2016 and Newton et al., 2015).

Reviewer #2: No minor issues, the data are beautifully presented and the text is well written.

PLOS authors have the option to publish the peer review history of their article (what does this mean?). If published, this will include your full peer review and any attached files.

Reviewer #1: No

Reviewer #2: No
---

## [Editor Report · Decision Letter 1]

31 Jan 2023

Dear Dr. Starai,

We are pleased to inform you that your manuscript 'Wbm0076, a candidate effector protein of the Wolbachia endosymbiont of Brugia malayi, disrupts eukaryotic actin dynamics' has been provisionally accepted for publication in PLOS Pathogens.

Best regards,

Edward Mitre

Academic Editor

PLOS Pathogens

James Collins III

Section Editor

PLOS Pathogens

Kasturi Haldar

Editor-in-Chief

PLOS Pathogens

orcid.org/0000-0001-5065-158X

Michael Malim

Editor-in-Chief

PLOS Pathogens

orcid.org/0000-0002-7699-2064

The authors have done a good job responding to reviewer critiques and, where necessary, clarifying the manuscript. The mansucript is well written, the figures are clear, and the findings add substantially to our understanding of the effects Wolbachia has on host cytoskeleton.
---

## [Editor Report · Acceptance letter]

13 Feb 2023

Dear Dr. Starai,

We are delighted to inform you that your manuscript, "Wbm0076, a candidate effector protein of the *Wolbachia* endosymbiont of *Brugia malayi*, disrupts eukaryotic actin dynamics," has been formally accepted for publication in PLOS Pathogens.

Best regards,

Kasturi Haldar

Editor-in-Chief

PLOS Pathogens

orcid.org/0000-0001-5065-158X

Michael Malim

Editor-in-Chief

PLOS Pathogens

orcid.org/0000-0002-7699-2064